# RegExplainer: Generating Explanations for Graph Neural Networks in Regression Tasks

Jiaxing Zhang[1], Zhuomin Chen[2], Hao Mei[3], Longchao Da[3], Dongsheng Luo[2], Hua Wei[3]

[1]New Jersey Institute of Technology, [2]Florida International University, [3]Arizona State University

[1]jz48@njit.edu, [2]{zchen051, dluo}@fiu.edu, [3]{hmei7, longchao, hua.wei}@asu.edu

## Abstract

Graph regression is a fundamental task that has gained significant attention in various graph learning tasks. However, the inference process is often not easily interpretable. Current explanation techniques are limited to understanding Graph Neural Network (GNN) behaviors in classification tasks, leaving an explanation gap for graph regression models. In this work, we propose a novel explanation method to interpret the graph regression models (XAIG-R). Our method addresses the distribution shifting problem and continuously ordered decision boundary issues that hinder existing methods away from being applied in regression tasks. We introduce a novel objective based on the graph information bottleneck theory (GIB) and a new mix-up framework, which can support various GNNs and explainers in a model-agnostic manner. Additionally, we present a self-supervised learning strategy to tackle the continuously ordered labels in regression tasks. We evaluate our proposed method on three benchmark datasets and a real-life dataset introduced by us, and extensive experiments demonstrate its effectiveness in interpreting GNN models in regression tasks.

## 1 Introduction

Graph Neural Networks [1] (GNNs) have become a powerful tool for learning knowledge from graph-structure data and achieved remarkable performance in many areas, including social networks [2, 3], molecular structures [4, 5], traffic flows [6–9], and recommendation systems [10–12]. Despite the success, their popularity in sensitive fields such as fraud detection and drug discovery [13, 14] requires an understanding of their decision-making processes. To address this challenge, some efforts have been made to explain GNN's predictions in a post-hoc manner, which aims to find a sub-graph that preserves the information about the predicted label. On top of the intuitive principle, Graph Information Bottleneck (GIB) [15, 16] maximizes the mutual information $I(G^*; Y)$ between the target prediction label $Y$ and the explanation $G^*$ while constraining the size of the explanation.

However, existing methods focus on the explanation of the classification tasks, leaving another fundamental task, explainable regression, unexplored. Graph regression tasks exist widely in nowadays applications, such as predicting the molecular property [17] or traffic flow volume [18]. Explaining the instance-level predictions of graph regression is challenging due to two main obstacles. First, in the routinely adopted GIB framework, the mutual information between the explanation sub-graph and label, $I(G^*; Y)$, is estimated with the Cross-Entropy between the predictions $f(G^*)$ from GNN model $f$ and its prediction label $Y$. However, in the regression task, the regression label is the continuous value, making the approximation unsuitable. Another challenge is the distribution shifting problem in the usage of $f(G^*)$, where the prediction of the explanation sub-graph made by the GNN model $f$ is unsafe. Usually, explanation sub-graphs have different topology and feature information compared to the original graph. As a result, explanation sub-graphs are out-of-distribution of the

original training graph dataset [19–21]. As shown in Figure 1, a GNN model $f$ is trained on the original graph training set and cannot be safely used to make predictions for sub-graphs.

To fill the gap, in this paper, we propose RegExplainer, to generate post-hoc instance-level explanations for graph regression tasks. Specifically, we formulate a theoretical-sound objective for explainable regression based on information theory. To further address the distribution shifting issue, RegExplainer develops a new mix-up approach with self-supervised learning. Our experiments show that RegExplainer provides consistent and concise explanations of GNN's predictions on regression tasks. We achieved up to $48.0\%$ improvement when compared to the alternative baselines in our experiments. Our contributions can be summarized as follows.

Figure 1: Intuitive illustration of the distribution shifting problem. The 3-dimensional map represents a trained GNN model $f$, where $(h_1, h_2)$ represents the embedding distribution of the graph in two dimensions, and $Y$ represents the prediction value of the graph through $f$. The red and blue lines represent the distribution of the original training graph set and the corresponding explanation sub-graph set, respectively. The distribution of $G^*$ shifts away from the original distribution, resulting in shifted prediction values.

• To our best knowledge, we are the first to explain GNN predictions on graph regression tasks. We addressed two challenges in explaining the graph regression task: the mutual information estimation in the GIB objective and the distribution shifting problem with continuous decision boundaries.

• We proposed a novel model with self-supervised learning and the mix-up approach, which can address the two challenges more effectively, and better explain the graph model on the regression tasks compared to other baselines.

• We designed three synthetic datasets, namely BA-Motif-Volume, BA-Motif-Counting and Triangles, as well as a real-world dataset called Crippen, which can also be used in future works, to evaluate the effectiveness of our regression task explanations. Comprehensive empirical studies on both synthetic and real-world datasets demonstrate that our method can provide consistent and concise explanations for graph regression tasks.

## 2 Related Work and Further Discussions

**GNN Explainability**   The explanation methods for GNN models can be categorized into two types based on their granularity: instance-level [22–25] and model-level [26], where the former methods explain the prediction for each instance by identifying important sub-graphs, and the latter method aims to understand the global decision rules captured by the GNN. These methods can also be classified into two categories based on their methodology: self-explainable GNNs [27, 28] and post-hoc explanation methods [23–25], where the former methods provide both predictions and explanations, while the latter methods use an additional model or strategy to explain the target GNN. Additionally, CGE [29] (cooperative explanation) generates the sub-graph explanation with the sub-network simultaneously, by using cooperative learning. However, it has to treat the GNN model as a white box, which is usually unavailable in the post-hoc explanation. Existing methods have only partially addressed the explanation of graph regression tasks and have not fully considered two important challenges: the distribution shifting problem and the limitations of the GIB objective, both of which are key areas our work aims to tackle.

**GIB Objective**   The Information Bottleneck (IB) [30, 31] provides an intuitive principle for learning dense representations that an optimal representation should contain *sufficient* information for the downstream prediction task with a *minimal* size. Based on IB, a recent work [32] unifies the most existing post-hoc explanation methods for GNN, such as GNNExplainer [23], PGExplainer [24], with the graph information bottleneck (GIB) principle [15, 16, 32]. Formally, the objective of explaining

the prediction of $f$ on $G$ can be represented by

$$\underset{G^*}{\arg\min} \, I(G; G^*) - \alpha I(G^*; Y), \tag{1}$$

where $G$ is the to-be-explained original graph, $G^*$ is the explanation sub-graph of $G$, $Y$ is the original ground-truth label of $G$, and $\alpha$ is a hyper-parameter to get the trade-off between minimal and sufficient constraints. GIB uses the mutual information $I(G; G^*)$ to select the minimal explanation that inherits only the most indicative information from $G$ to predict the label $Y$ by maximizing $I(G^*; Y)$, where $I(G; G^*)$ avoids imposing potentially biased constraints, such as the size or the connectivity of the selected sub-graphs [15]. Through the optimization of the sub-graph, $G^*$ provides model interpretation. In graph classification task, a widely-adopted approximation to Eq. (1) in previous methods [23, 24] is:

$$\underset{G^*}{\arg\min} \, I(G; G^*) + \alpha H(Y|G^*) \approx \underset{G^*}{\arg\min} \, I(G; G^*) + \alpha \mathrm{CE}(Y, Y^*),$$

where $Y$ and $Y^*$, approximated by $f(G)$ and $f(G^*)$, is the predicted label of $G$ and $G^*$ made by the to-be-explained model $f$, and the cross-entropy $\mathrm{CE}(Y, Y^*)$ between $Y$ and $Y^*$ is used to approximate $-I(G^*; Y)$. The approximation is based on the definition of mutual information $I(G^*; Y) = H(Y) - H(Y|G^*)$: with entropy $H(Y)$ being static and independent of the explanation process, minimizing the mutual information between the explanation sub-graph $G^*$ and $Y$ can be reformulated as maximizing the conditional entropy of $Y$ given $G^*$, which can be approximated by $\mathrm{CE}(Y, Y^*)$.

## 3 Preliminary

**Notation and Problem Formulation** We use $G = (\mathcal{V}, \mathcal{E}; \boldsymbol{X}, \boldsymbol{A})$ to represent a graph from an alphabet $\mathcal{G}$, where $\mathcal{V}$ equals to $\{v_1, v_2, ..., v_n\}$ represents a set of $n$ nodes and $\mathcal{E} \in \mathcal{V} \times \mathcal{V}$ represents the edge set. Each graph has a feature matrix $\boldsymbol{X} \in \mathbb{R}^{n \times d}$ for the nodes, wherein $\boldsymbol{X}, X_i \in \mathbb{R}^{1 \times d}$ is the $d$-dimensional node feature of node $v_i$. $\mathcal{E}$ is described by an adjacency matrix $\boldsymbol{A} \in \{0, 1\}^{n \times n}$, where $A_{ij} = 1$ means that there is an edge between node $v_i$ and $v_j$; otherwise, $A_{ij} = 0$. For the graph prediction task, each graph $G_k$ has a label $Y_k \in \mathcal{C}$, where $k \in \{1, ..., N\}$, $N$ represents the number of graphs in the dataset, $\mathcal{C}$ is the set of the classification categories or regression values in $\mathbb{R}$, with a GNN model $f$ trained to make the prediction, i.e., $f : (\boldsymbol{X}, \boldsymbol{A}) \mapsto \mathcal{C}$.

**Problem 1** (Post-hoc Instance-level GNN Explanation). *Given a trained GNN model $f$, for an arbitrary input graph $G = (\mathcal{V}, \mathcal{E}; \boldsymbol{X}, \boldsymbol{A})$, the goal of post-hoc instance-level GNN explanation is to find a sub-graph $G^*$ that can explain the prediction of $f$ on $G$.*

In non-graph structured data, the informative feature selection has been well studied [33], as well as in traditional methods, such as concrete auto-encoder [34], which can be directly extended to explain features in GNNs. In this paper, we focus on discovering the important sub-graph typologies following the previous work [23, 24]. Specifically, the obtained explanation $G^*$ is depicted by a binary mask $\boldsymbol{M}^* \in \{0, 1\}^{n \times n}$ on the adjacency matrix, e.g., $G^* = (\mathcal{V}, \mathcal{E}; \boldsymbol{X}, \boldsymbol{A} \odot \boldsymbol{M}^*)$, $\odot$ means elements-wise multiplication. The mask highlights components of $G$ which are essential for $f$ to make the prediction.

## 4 Methodology

In this section, we first introduce a new objective based on GIB for explaining graph regression tasks. Then we showcase the distribution shifting problem in the objective for regression and propose a novel framework with the mix-up approach to solve the distribution shifting problem, by incorporating the mix-up approach with self-supervised contrastive learning.

### 4.1 GIB for Explaining Graph Regression

As introduced in Section 2, in the classification task, $I(G^*; Y)$ in Eq. (1) is commonly approximated by cross-entropy $\mathrm{CE}(Y^*, Y)$ [35]. However, it is non-trivial to extend it for regression tasks because $Y$ is a continuous variable and it is intractable to compute the cross-entropy $\mathrm{CE}(Y^*, Y)$ or the mutual information $I(G^*; Y)$, where $G^*$ is a graph variable with a continuous variable $Y^*$ as its label.

### 4.1.1 Optimizing the Lower Bound of $I(G^*; Y)$

To address the challenge of computing the mutual information $I(G^*; Y)$ with a continuous $Y$, we propose a novel objective for explaining graph regression.

Instead of minimizing $I(G^*; Y)$ directly, we propose to maximize a lower bound for the mutual information by including the prediction label of $G^*$, denoted by $Y^*$, and approximate $I(G^*; Y)$ in Eq. (1) with $I(Y^*; Y)$:

$$\arg\min_{G^*} I(G; G^*) - \alpha I(Y^*; Y). \tag{2}$$

$I(Y^*; Y)$ has the following property, upon which we can approximate Eq. (2):

**Property 1** $I(Y^*; Y)$ *is a lower bound of* $I(G^*; Y)$.

Intuitively, the property of $I(Y^*; Y)$ is guaranteed by the chain rule for mutual information and the independence between each explanation instance $g^*$ in $G^*$. An intuitive demonstration is shown in Figure 2. The proof is shown in the Appendix B.1.

### 4.1.2 Estimating $I(Y^*; Y)$ with InfoNCE

Now the challenge becomes the estimation of the mutual information $I(Y^*; Y)$. Inspired by the model of Contrastive Predictive Coding [36], in which InfoNCE loss is interpreted as a mutual information estimator, we further adapt the objective function so that it can be applied with

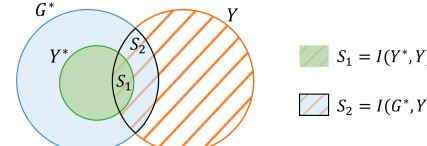

Figure 2: Intuitive illustration about why $I(G^*; Y) \geq I(Y^*; Y)$. $G^*$ contains more mutual information as having more overlapping area with $Y$ than the overlapping area between $Y^*$ and $Y$.

InfoNCE loss in explaining graph regression. In our graph explanation scenario, the InfoNCE Loss defined in Eq. (3) can also be utilized as a lower bound of $I(Y^*; Y)$, as shown in the following property with proofs:

**Property 2** InfoNCE Loss is a lower bound of the $I(Y^*; Y)$:

$$I(Y^*; Y) \geq \mathop{\mathbb{E}}_{\mathbb{Y}} \left[ \log \frac{\text{sim}(Y^*, Y)}{\frac{1}{|\mathbb{Y}|} \sum_{Y' \in \mathbb{Y}} \text{sim}(Y^*, Y')} \right], \tag{3}$$

where $Y'$ is the prediction label of the randomly sampled graph neighbors, $\mathbb{Y}$ is the set of the neighbors' prediction labels, and $\text{sim}()$ estimates the similarity between $Y^*$ and $Y$. The proof is shown in the Appendix B.2. Therefore, we have the InfoNCE loss $\mathcal{L}_{\text{NCE}}$ as the lower bound of the $I(Y^*; Y)$. We approximate Eq. (2) as:

$$\arg\min_{G^*} I(G; G^*) - \alpha \mathop{\mathbb{E}}_{\mathbb{Y}} \left[ \log \frac{\text{sim}(Y^*, Y)}{\frac{1}{|\mathbb{Y}|} \sum_{Y' \in \mathbb{Y}} \text{sim}(Y^*, Y')} \right]. \tag{4}$$

## 4.2 Distribution Shifting Problem in Graph Regression

We include the prediction label $Y^*$ in Eq. (4) to estimate similarity, which is approximated with $f(G^*)$ in previous work [23, 24]. However, we argue that $f(G^*)$ cannot be safely obtained due to the distribution shift problem [37, 19]. In classification tasks, a small shift may not cross the decision boundaries, which can still lead to a correct prediction. However, due to the continuous decision boundaries in regression, the distribution problem would cause serious prediction errors. Here in this paper, the graph distribution is indicated by its regression label in the regression task.

Figure 3 shows the existence of distribution shifts between $f(G^*)$ and $f(G)$ in graph regression tasks. For each dataset, we sort the indices of the data samples according to the value of their labels, and visualize the label $Y$, prediction $f(G)$ of the original graph from the trained GNN model $f$, and prediction $f(G^*)$ of the explanation sub-graph $G^*$ from $f$. As we can see in Figure 3, in all four graph regression datasets, the red points are well distributed around the ground-truth blue points, indicating that $f(G)$ is close to $Y$. In comparison, the green points shift away from the red points, indicating the shifts between $f(G^*)$ and $f(G)$. Especially in dataset BA-Motif-Counting, the sub-graph explanation distribution was shifted extremely.

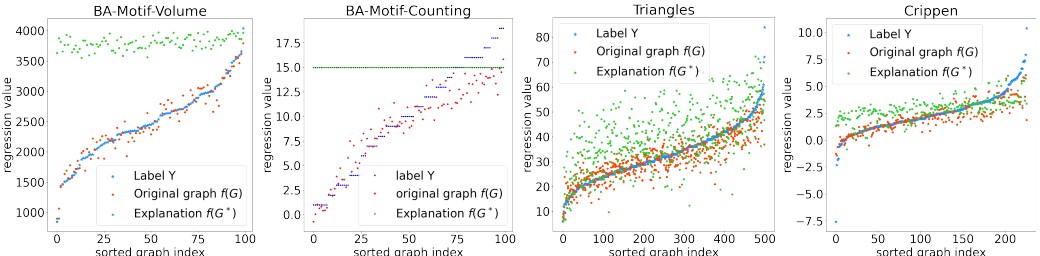

Figure 3: Visualization of distribution shifting problem on four graph regression datasets. The points represent the regression value, where the blue points mean ground truth label $Y$, red points mean prediction $f(G)$, and the green points mean prediction $f(G^*)$ on the four datasets. The x-axis is the indices of the graph, sorted by the value of the label $Y$.

Intuitively, this phenomenon indicates the GNN model $f$ can make correct predictions only with the original graph $G$ yet can not predict the explanation sub-graph $G^*$ correctly. This is because the GNN model $f$ is trained with the original graph sets, whereas the explanation $G^*$ as the sub-graph is different from the original graph sets. With the shift between $f(G)$ and $f(G^*)$, the optimal solution in Eq. (4) is unlikely to work well.

## 4.3 Mix-up Approach with Contrastive Learning

To address this distribution-shifting problem in graph regression, we innovatively incorporate the mix-up approach with a self-supervised contrastive learning strategy. Instead of calculating $Y^*$ with $f(G^*)$ directly, we approximate with $Y^{(\mathrm{mix})}$ from $f(G^{(\mathrm{mix})})$, which contains similar information as $G^*$ but is in the same distribution with $G$. Specifically, our approach includes the following steps:

• **Step 1 (Neighbor Sampling):** Learning through the triplet instances can effectively reinforce the ability of the explainer to learn the explanation self-supervised. For each target graph $G$ with label $Y$ to be explained, we can define two randomly sampled graphs as positive neighbor $G^+$ and negative neighbor $G^-$, where $G^+$'s label $Y^+$ is closer to $Y$ than $G^-$'s label $Y^-$, i.e., $|Y^+ - Y| < |Y^- - Y|$. Intuitively, the distance between the distributions of the positive pair $\langle G, G^+ \rangle$ should be smaller than the distance between the distributions of the negative pair $\langle G, G^- \rangle$.

• **Step 2 (Mixup for $G^*$):** Then we generate two mixup graphs $G^{(\mathrm{mix})+}$ and $G^{(\mathrm{mix})-}$ by mixing the sub-graph explanation $G^*$ with the label irrelevant sub-graph $(G^+)^\Delta = G^+ - (G^+)^*$ from its positive neighbor $G^+$ and the label irrelevant sub-graph $(G^-)^\Delta = G^- - (G^-)^*$ from negative neighbor $G^-$ respectively. Specifically, the label mixup approach is calculated from:

$$G^{(\mathrm{mix})+} = G^* + (G^+)^\Delta = G^* + (G^+ - (G^+)^*), G^{(\mathrm{mix})-} = G^* + (G^-)^\Delta = G^* + (G^- - (G^-)^*).$$

$G^{(\mathrm{mix})+}$ and $G^{(\mathrm{mix})-}$ should have the similar information to $G$ because they have the same label-preserving sub-graphs $G^*$. Additionally, considering the following two pairs: $(G^{(\mathrm{mix})+}, G^+)$ and $(G^{(\mathrm{mix})-}, G^-)$. The similarity between $(G^{(\mathrm{mix})+}, G^+)$ should be larger than the similarity between $(G^{(\mathrm{mix})-}, G^-)$. Intuitively, since $G^{(\mathrm{mix})+}$ and $G^{(\mathrm{mix})-}$ have the same label-preserving sub-graphs $G^*$ and $|Y^- - Y| > |Y^+ - Y|$, we can have $|f(G^-) - f(G^{(\mathrm{mix})-})| > |f(G^+) - f(G^{(\mathrm{mix})+})|$, where $f(G)$ represents the prediction label of graph $G$.

• **Step 3 (InfoNCE Loss Approximation):** Then we can safely estimate the similarity with $\mathrm{sim}(Y^{(\mathrm{mix})}, Y)$. To save more information, we use the similarity of representation embedding to approximate the similarity of the graph prediction label, where $\boldsymbol{h}^{(\mathrm{mix})}$ represents the embedding for $G^{(\mathrm{mix})}$ and $\boldsymbol{h}$ represents the embedding for $G$. We use $\mathbb{H}$ to represent the neighbors set accordingly. Thus, we approximate Eq. (4) as:

$$\underset{G^*}{\arg\min}\, I(G; G^*) - \alpha \underset{\mathbb{H}}{\mathbb{E}} \left[ \log \frac{\mathrm{sim}\left(\boldsymbol{h}^{(\mathrm{mix})}, \boldsymbol{h}\right)}{\frac{1}{|\mathbb{H}|} \sum_{\boldsymbol{h}' \in \mathbb{H}} \mathrm{sim}\left(\boldsymbol{h}^{(\mathrm{mix})}, \boldsymbol{h}'\right)} \right]. \tag{5}$$

**Different between Mix-up Approach in Classification Tasks** The mix-up approach in previous work [19] generates a mixed graph by simply mixing explanation sub-graph $G^*$ with a randomly sampled label-irrelevant sub-graph $G^\Delta$ [19], which can be formally written as $G_a^{(\mathrm{mix})} = G_a^* + (G_b -$

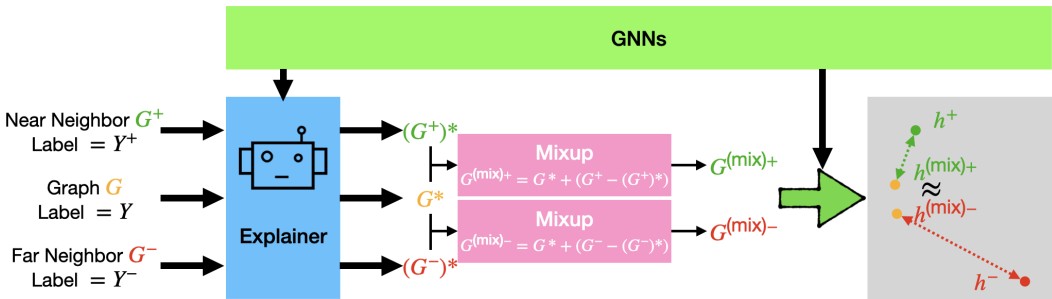

Figure 4: Illustration of RegExplainer. $G$ is the to-be-explained graph, $G^+$ and $G^-$ are the randomly sampled positive and negative neighbors. The explanation of the graph is produced by the explainer model. Then graph $G^*$ is mixed with $(G^+)^\Delta = G^+ - (G^+)^*$ and $(G^-)^\Delta = G^- - (G^-)^*$ respectively to produce $G^{(\text{mix})+}$ and $G^{(\text{mix})-}$. Then the graphs are fed into the trained GNN model to retrieve the embedding vectors $\boldsymbol{h}^+$, $\boldsymbol{h}^-$, $\boldsymbol{h}^{(\text{mix})+}$ and $\boldsymbol{h}^{(\text{mix})-}$, where $\boldsymbol{h}^{(\text{mix})+} \approx \boldsymbol{h}^{(\text{mix})-}$ due to the same label-preserving sub-graph $G^*$. We use InfoNCE loss to minimize the distance between $G^{(\text{mix})+}$ and the positive sample and maximize the distance between $G^{(\text{mix})-}$ and the negative sample. The explainer is trained with the GIB objective and self-supervised contrastive loss.

$G_b^*$). However, it cann't tackle the continuous decision boundaries in graph regression tasks. A detailed description of the mix-up approach can be found in Appendix C.

### 4.4 Implementation

**InfoNCE Loss**  After generating the mix-up explanation $G^{(\text{mix})}$, we specify the InfoNCE loss to further train the parameterized explainer with a triplet of graphs $\langle G, G^+, G^- \rangle$. In practice, $G^+$ and $G^-$ are randomly sampled from the graph dataset, upon which we calculate their similarity score with the target graph $G$. The sample with a higher score would be the positive sample and the other one would be the negative sample. Specifically, we use $\text{sim}(\boldsymbol{h}, \boldsymbol{h}') = \boldsymbol{h}^\top \boldsymbol{h}'$ to compute the similarity score, where $G'$ can be $G^+$ or $G^-$. $\boldsymbol{h}$ is generated by feeding $G$ into the GNN model $f$ and retrieving the embedding vector before the dense layers.

Formally, given a target graph $G$, the sampled positive graph $G^+$ and negative graph $G^-$, we formulate the InfoNCE loss in Eq. (5) as the following:

$$\mathcal{L}_{\text{NCE}}(G, G^+, G^-) = -\log \frac{\exp((\boldsymbol{h}^{(\text{mix})+})^\top \boldsymbol{h})}{\exp((\boldsymbol{h}^{(\text{mix})+})^\top \boldsymbol{h}^+) + \exp((\boldsymbol{h}^{(\text{mix})-})^\top \boldsymbol{h}^-)}, \tag{6}$$

where $\exp(\boldsymbol{h}^\top \boldsymbol{h})$ is used to instantiate the function sim, the denominator is a sum over the similarities of both positive and negative samples.

**Size Constraints**  We optimize $I(G; G^*)$ in Eq. (5) to constraint the size of the explanation sub-graph $G^*$. The upper bound of $I(G; G^*)$ is optimized as the estimation of the KL-divergence between the probabilistic distribution between the $G^*$ and $G$, where the KL-divergence term can be divided into two parts as the entropy loss and size loss [16]. In practice, we follow the previous work [23, 24, 38] to implement them. Specifically,

$$\mathcal{L}_{\text{size}}(G, G^*) = \gamma \sum_{(i,j) \in \mathcal{E}} (M_{ij}^*) - \log \sigma((\boldsymbol{h}^*)^\top \boldsymbol{h}^*), \tag{7}$$

where $\sum_{(i,j) \in \mathcal{E}} (M_{ij}^*)$ means sum the weights of the existing edges in the edge weight mask $\boldsymbol{M}^*$ for the explanation $G^*$; $\boldsymbol{h}^*$ is extracted from the embedding of the graph $G^*$ before the GNN model $f$ transforming it into prediction $Y^*$, $\sigma$ means the sigmoid function and $\gamma$ is the weight for the size of the masked graph. In implementation, we set $\gamma = (0.0003, 0.3)$ following previous work [19].

**Overall Objective Function**  In practice, the denominator in Eq. (5) works as a regularization to avoid trivial solutions. Since the label $Y$ is given and independent of the optimization process, we can also employ the MSE loss between $Y^*$ and $Y$ additionally, regarding InfoNCE loss only estimates the

Table 1: Illustration of the graph regression datasets together with the explanation faithfulness in terms of AUC-ROC on edges under four datasets on RegExplainer and other baselines. The original graph row visualizes the structure of the complete graph, the explanation row highlights the explanation sub-graph of the corresponding original graph. In the Crippen dataset, different colors of the node represent different kinds of atoms and the node feature is a one-hot vector to encode the atom type.

| Dataset | BA-Motif-Volume | BA-Motif-Counting | Triangles | Crippen |
|---|---|---|---|---|
| Original Graph $G$ |  |  |  |  |
| Explanation $G^*$ | | | | |
| Node Feature | Random Float Vector | Fixed Ones Vector | Fixed Ones Vector | One-hot Vector |
| Regression Label | Sum of Motif Value | Number of Motifs | Number of Triangles | Chemical Property Value |
| Explanation Type | Fix Size Sub-Graph | Dynamic Size Sub-graph | Dynamic Size Sub-graph | Dynamic Size Sub-graph |
| Explanation AUC | | | | |
| GRAD | $0.418 \pm 0.000$ | $0.527 \pm 0.000$ | $0.479 \pm 0.000$ | $0.426 \pm 0.000$ |
| ATT | $0.512 \pm 0.005$ | $0.521 \pm 0.003$ | $0.441 \pm 0.004$ | $0.502 \pm 0.006$ |
| MixupExplainer | $0.471 \pm 0.0291$ | $0.868 \pm 0.127$ | $0.663 \pm 0.110$ | $0.499 \pm 0.002$ |
| GNNExplainer | $0.501 \pm 0.009$ | $0.505 \pm 0.004$ | $0.500 \pm 0.002$ | $0.497 \pm 0.005$ |
| **+RegExplainer** | $0.588 \pm 0.017$ | $0.629 \pm 0.001$ | $0.537 \pm 0.003$ | $0.541 \pm 0.011$ |
| PGExplainer | $0.470 \pm 0.057$ | $0.798 \pm 0.133$ | $0.511 \pm 0.028$ | $0.448 \pm 0.005$ |
| **+RegExplainer** | $\mathbf{0.758 \pm 0.177}$ | $\mathbf{0.989 \pm 0.003}$ | $\mathbf{0.739 \pm 0.008}$ | $\mathbf{0.553 \pm 0.013}$ |

mutual information between the embeddings. Formally, the overall loss function can be implemented as:

$$\mathcal{L} = \mathcal{L}_{\text{GIB}} + \beta \mathcal{L}_{\text{MSE}}(f(G), f(G^{(\text{mix})+})), \text{where } \mathcal{L}_{\text{GIB}} = \mathcal{L}_{\text{size}}(G, G^*) - \alpha \mathcal{L}_{\text{NCE}}(G, G^+, G^-) \quad (8)$$

$G^{(\text{mix})+}$ means mix $G^*$ with the positive sample $G^+$ and $\alpha$ and $\beta$ are hyper-parameters. The training algorithm and description of it are put in Appendix D.

## 5 Experiments

In this section, we conduct experiments to demonstrate the performance of our proposed method[1]. These experiments are mainly designed to explore the following research questions:

• **RQ1:** Can RegExplainer outperforms other baselines in explaining GNNs on regression tasks?

• **RQ2:** How does each part of RegExplainer and hyperparameters impact the overall performance in generating explanations?

• **RQ3:** Does the distribution shifting exist in GNN explanation? Can RegExplainer alleviate it?

### 5.1 Experiment Settings

We formulate Three synthetic datasets and a real-world dataset, as is shown in Table 1, in order to address the lack of graph regression datasets with ground-truth explanation. The datasets include: **BA-Motif-Volume** and **BA-Motif-Counting**, which are based on BA-shapes [23], **Triangles** [39], and **Crippen** [40]. We compared the proposed RegExplainer against a comprehensive set of baselines in all datasets, including: **GRAD** [23], **ATT** [41], **GNNExplainer** [23], **PGExplainer** [24], and **MixupExplainer** [19]. Detailed information about experiment setting are put in the Appendix E. We elaborate on the measurement metric of methods as follows: (1) **AUC-ROC**: We use the AUC score to evaluate the performance of our proposed methods against baseline methods regarding the ground-truth explanation, which can be treated as a binary classification task. (2) We evaluate the similarity of distribution of the graph with **Cosine Similarity** and **Euclidean Distance**.

### 5.2 Quantitative Evaluation (RQ1)

In this section, we evaluate the performance of our approach with other baselines. For GRAD and GAT, we use the gradient-based and attention-based explanation, following the setting in the

---

[1]Our data and code are available at: `https://github.com/jz48/RegExplainer`

previous work [23]. We take GCN as our to-be-explained model for all post-hoc explainers. For GNNExplainer, PGExplainer, and MixupExplainer, which were previously used for the classification task, we replace the Cross-Entropy loss with the MSE loss. We run and tune all the baselines on our four datasets. We evaluate the explanation from all the methods with the AUC metric, as done in the previous work. As we can see in Table 1, we take GNNExplainer and PGExplainer as backbones and apply our framework as RegExplainer on both of them. The experiment results demonstrate the effectiveness of our methods in explaining graph regression tasks, where our method achieves the best performance compared to the baselines in all four datasets.

In Table 1, RegExplainer based on PGExplainer improves the second best baseline with $0.175/34.3\%$ on average and up to $0.246/48.0\%$. The comparison between RegExplainer and other baselines indicates the advantages of our proposed approach. This improvement indicates the effectiveness of our proposed method, showing that by incorporating the mix-up approach and contrastive learning, we can generate more faithful explanations in the graph regression tasks. In the following sections, we analyze the RegExplainer with PGExplainer as a backbone.

### 5.3 Ablation Study and Hyper-parameter Sensitivity Study (RQ2)

We conducted an ablation study to show how our proposed components, specifically, the mix-up approach and self-supervised learning, contribute to the final performance of RegExplainer. To this end, we denote RegExplainer as RegE and design three types of variants as follows: (1) RegE$^{-\text{mix}}$: We remove the mix-up processing after generating the explanations and feed the sub-graph $G^*$ into the objective function directly. (2) RegE$^{-\text{nce}}$: We remove the InfoNCE loss term but still maintain the mix-up processing and MSE loss. (3) RegE$^{-\text{mse}}$: We remove the MSE loss computation item from the objective function.

Figure 5: Ablation study of RegExplainer. We evaluated the AUC performance of the original RegExplainer and its variants that exclude the mix-up approach, InfoNCE loss, or MSE loss, respectively. The black solid line shows the standard deviation.

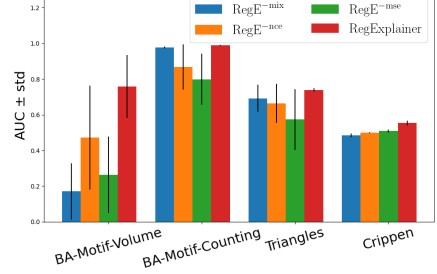

Additionally, we set all variants with the same configurations as original RegExplainer, including learning rate, training epochs, and hyper-parameters $\eta$, $\alpha$, and $\beta$. We trained them on all four datasets and conducted the results in Figure 5. We observed that the proposed RegExplainer outperforms its variants in all datasets, which indicates that each component is necessary and the combination of them is effective.

We also investigate the hyper-parameters of our approach, which include $\alpha$ and $\beta$, across all four datasets. The hyper-parameter $\alpha$ controls the weight of the InfoNCE loss in the GIB objective while the $\beta$ controls the weight of the MSE loss. We determined the optimal values of $\alpha$ and $\beta$ with grid search. The experimental results can be found in Figure 6. We fixed $\alpha$ and $\beta$ at 1 and changed another parameter to visualize the change in model performance. Figure 6 illustrates that the model's performance is robust to changes in hyper parameters within the scope $[0.001, 1000]$. Our findings indicate that our approach, RegExplainer, is stable and robust when using different hyper-parameter settings, as evidenced by consistent performance across a range.

Figure 6: Hyper-parameters study of $\alpha$ and $\beta$ on four datasets with RegExplainer. In both figures, the x-axis is the value of different hyper-parameter settings, and the y-axis is the value of the average AUC score over ten runs with different random seeds.

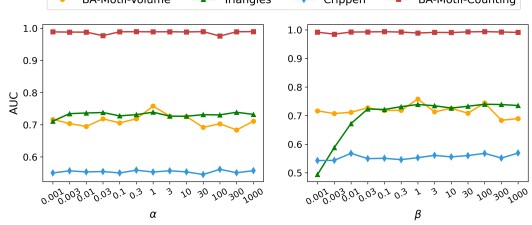

Table 2: Prediction shifting study on the RMSE of $(f(G), Y)$, $(f(G^*), Y)$, $(f(G), f(G^*))$ respectively.

| Dataset | $(f(G), Y)$ | $(f(G^*), Y)$ | $(f(G), f(G^*))$ |
|---|---|---|---|
| BA-Motif-Volume | 131.42 | 1432.07 | 1427.07 |
| BA-Motif-Counting | 2.06 | 7.43 | 7.22 |
| Triangles | 5.28 | 12.38 | 12.40 |
| Crippen | 1.13 | 1.54 | 1.17 |

## 5.4 Alleviating Distribution Shifts (RQ3)

In this section, we visualize the regression values of the graphs and calculate the prediction shifting distance for each dataset and analyze their correlations to the distance of the decision boundaries. We put our results into Figure 7 and Table 2.

We observed that in Figure 3, red points surround the blue points but green points are shifted away, which indicates that the explanation sub-graph cann't help GNNs make correct predictions. As shown in Table 2, we calculate the RMSE score between the $f(G)$ and $Y$, $f(G^*)$ and $Y$, $f(G)$ and $f(G^*)$ respectively, where $f(G)$ is the prediction the original graph, $f(G^*$ is the prediction of the explanation sub-graph, and $Y$ is the regression label. We can observe that $f(G^*)$ shows a significant prediction shifting from $f(G)$ and $Y$, indicating that the mutual information calculated with $(f(G^*), Y)$ would be biased.

We further explore the relationship of the prediction shifting against the label value with dataset BA-Motif-Volume, which represents the semantic decision boundary. This additional experiment with Figure 7 can be found in Appendix F.1.

We also design experiments to illustrate how RegExplainer corrects the deviations: we calculate the graph embeddings $v$ and predictions $p$ of the explanation sub-graphs and the mix-up graph. Then we compare them to the ground truth and calculate the Euclidean or Cosine distance between the vectors and RMSE between prediction labels. From the results in Table 3, we can observe that all the performances of $COS(v_g, v_m)$, $EUC(v_g, v_m)$ and prediction errors are better than those of $(v_g, v_e)$, which indicates RegExplainer can effectively fix the distribution of sub-graph explanation $G^*$ and reduce the embedding distance and prediction error.

Table 3: Table for measuring distribution repairing. $v_g$, $v_e$ and $v_m$ are the embeddings from $f$ of original graph $G$, explanation subgraph $G^*$ and the mix-up explanation $G^{(\text{mix})+}$. $p_g$, $p_e$ and $p_m$ are the predicted labels for the original graph, explanation subgraph and the mix-up explanation. EUC means Euclidean distance ($\downarrow$, the smaller the better) and COS means cosine distance ($\uparrow$, the larger the better). RMSE means Root Mean Square Error ($\downarrow$, the smaller the better).

| | BA-Motif-Volume | BA-Motif-Counting | Triangles | Crippen |
|---|---|---|---|---|
| $COS(v_g, v_e)$ | 0.95 | 0.80 | 0.97 | 0.89 |
| $COS(v_g, v_m)$ | 0.98 | 0.89 | 0.99 | 0.92 |
| $EUC(v_g, v_e)$ | 0.46 | 0.68 | 0.19 | 0.67 |
| $EUC(v_g, v_m)$ | 0.37 | 0.52 | 0.08 | 0.63 |
| $RMSE(p_g, p_e)$ | 1427.07 | 7.22 | 12.40 | 1.17 |
| $RMSE(p_g, p_m)$ | 393.26 | 2.73 | 8.22 | 0.68 |

## 6 Conclusion

We addressed the challenges in the explainability of graph regression tasks and proposed the RegExplainer, a novel method for explaining the predictions of GNNs with the post-hoc explanation sub-graph on graph regression task without requiring modification of the underlying GNN architecture or re-training. We showed how RegExplainer can leverage the mix-up approach to solve the distribution shifting problem and adopt the GIB objective with the InfoNCE loss to migrate it from graph classification tasks to graph regression tasks, while these existing challenges seriously affect the performances of other explainers. We formulated four new datasets: BA-Motif-Volume, BA-Motif-Counting, Triangles, and Crippen for evaluating the explainers on the graph regression

task, which are aligned with the design of datasets in previous work. They can also benefit future studies on the XAIG-R. While we acknowledge the effectiveness of our method, we also recognize its limitations. Specifically, although our approach can be applied to explainers for graph regression tasks in an explainer-agnostic manner, it cannot be easily applied to explainers built for explaining the spatio-temporal graph due to the dynamic topology and node features of the STG. To overcome this challenge, a potential solution is to incorporate cached dynamic embedding memories into the framework.

# 7 Ethics Statement

This work is primarily foundational in GNN explainability, focusing on expanding the GIB objective function of the explainer framework from graph classification tasks to graph regression tasks. Its primary aim is to contribute to the academic community by enhancing the explanation in graph regression. We do not foresee any direct, immediate, or negative societal impacts stemming from the outcomes of our research.

# 8 Acknowledgments

The work was partially supported by NSF awards #2421839 and #2331908. The views and conclusions contained in this paper are those of the authors and should not be interpreted as representing any funding agencies.

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

# A Symbols Table

| Symbol Name | Symbol Meaning |
|:---:|:---|
| $G$ | Original to-be-explained graph |
| $G^*$ | Optimized sub-graph explanation |
| $Y$ | Prediction label for $G$ |
| $Y^*$ | Prediction label for $G^*$ |
| $I(\cdot)$ | Mutual Information |
| $f(\cdot)$ | Prediction made by to-be-explained GNN model $f$ |
| $(h_1, h_2)$ | The 2-dims representation of graph embeddings |
| $\alpha$ | Hyper parameter for mutual information term in GIB |
| $H(\cdot)$ | Information entropy |
| $\mathcal{V}$ | Node set |
| $\mathcal{E}$ | Edge set |
| $\boldsymbol{X}$ | Feature matrix |
| $\boldsymbol{A}$ | Adjacency matrix |
| $\mathcal{G}$ | Graph set |
| $v$ | A node in graph |
| $v_i$ | The $i$-th node in graph |
| $n$ | Number of nodes |
| $d$ | Dimension of feature |
| $i$ | The $i$-th node index |
| $j$ | The $j$-th node index |
| $k$ | The $k$-th graph index |
| $A_{ij}$ | edge from node i to node j |
| $\mathcal{C}$ | A set of classification categories or $\mathbb{R}$ for regression tasks |
| $\boldsymbol{M}^*$ | Edge mask which denotes $G^*$ in $G$ |
| $\mathbb{Y}$ | Set of neighbors' prediction labels |
| $Y'$ | Prediction label of sampled graph neighbor |
| $\boldsymbol{h}$ | Graph embedding for $G$ |
| $\mathbb{H}$ | Set of $\boldsymbol{h}$ |
| $\gamma$ | Hyper parameter for leveraging sum of edge weights |
| $\sigma$ | Sigmoid function |
| $\beta$ | Hyper parameter for MSE loss |
| $v_g$ | Embedding vectors for graph $G$ |
| $v_e$ | Embedding vectors for explanation sub-graph $G^*$ |
| $v_m$ | Embedding vectors for mix-up graph $G^{(\text{mix})+}$ |
| $p_g$ | Prediction labels for graph $G$ |
| $p_e$ | Prediction labels for explanation sub-graph $G^*$ |
| $p_m$ | Prediction labels for mix-up graph $G^{(\text{mix})+}$ |
| $h(\cdot)$ | Mapping function from $G^*$ to $Y^*$ |
| $y^*$ | An instance of $Y^*$ |
| $g^*$ | An instance of $G^*$ |

Table 4: Important notations and symbols table.

# B Proof

## B.1 Property 1

*Proof.* From the definition of $Y^*$, we can make a safe assumption that there is a many-to-one map (function), denoted by $h$, from $G^*$ to $Y^*$ as $Y^*$ is the prediction label for $G^*$. For simplicity, we assume a finite number of explanation instances for each label $y^*$, and each explanation instance, denoted by $g^*$, is generated independently. Then, we have $p(y^*) = \sum_{g^* \in \mathbb{G}(y^*)} p(g^*)$, where $\mathbb{G}(y^*) = \{g|h(g) = y^*\}$ is the set of explanations whose labels are $y^*$.

Based on the definition of mutual information, we have:

$$I(G^*; Y) = \int_y \int_{g^*} p_{(G^*,Y)}(g^*, y) \log \frac{p_{(G^*,Y)}(g^*, y)}{p_{G^*}(g^*)p_Y(y)} d_{g^*} d_y$$

$$= \int_y \int_{g^*} p_{(G^*,Y^*,Y)}(g^*, h(g^*), y)$$

$$\log \frac{p_{(G^*,Y^*,Y)}(g^*, h(g^*), y)}{p_{G^*}(g^*)p_Y(y)} d_{g^*} d_y$$

$$= \int_y \int_{g^*} p_{(G^*,Y^*,Y)}(g^*, h(g^*), y)$$

$$\log \frac{p_{(G^*,Y^*,Y)}(g^*, h(g^*), y)}{p_{(G^*,Y^*)}(g^*, h(g^*))p_Y(y)} d_{g^*} d_y$$

$$= \int_y \int_{y^*} \sum_{g^* \in \mathbb{G}(y^*)} p_{(G^*,Y^*,Y)}(g^*, y^*, y)$$

$$\log \frac{p_{(G^*,Y^*,Y)}(g^*, y^*, y)}{p_{(G^*,Y^*)}(g^*, y^*)p_Y(y)} d_{y^*} d_y$$

Based on our many-to-one assumption, while each $g^*$ is generated independently, we know that if $g \notin \mathbb{G}(y^*)$, then we have $p_{(G^*,Y^*,Y)}(g^*, y^*, y) = 0$. Thus, we have:

$$I(G^*; Y) = I(G^*; Y)$$

$$+ \int_y \int_{y^*} \sum_{g \notin \mathbb{G}(y^*)} p_{(G^*,Y^*,Y)}(g^*, y^*, y)$$

$$\log \frac{p_{(G^*,Y^*,Y)}(g^*, y^*, y)}{p_{(G^*,Y^*)}(g^*, y^*)p_Y(y)} d_{y^*} d_y$$

$$= \int_y \int_{y^*} \int_{g*} p_{(G^*,Y^*,Y)}(g^*, y^*, y)$$

$$\log \frac{p_{(G^*,Y^*,Y)}(g^*, y^*, y)}{p_{(G^*,Y^*)}(g^*, y^*)p_Y(y)} d_{g^*} d_{y^*} d_y$$

$$= I(G^*, Y^*; Y).$$

With the chain rule for mutual information, we have $I(G^*, Y^*; Y) = I(Y^*; Y) + I(G^*; Y|Y^*)$. Then due to the non-negativity of the mutual information, we have $I(G^*, Y^*; Y) \geq I(Y^*; Y)$. $\square$

## B.2 Property 2

*Proof.* As in the InfoNCE method, the mutual information between $Y^*$ and $Y$ is defined as:

$$I(Y^*; Y) = \sum_{Y^*,Y} p(Y^*, Y) \log \frac{p(Y|Y^*)}{P(Y)} \tag{9}$$

However, the ground truth joint distribution $p(Y^*, Y)$ is not controllable, so, we turn to maximize the similarity

$$\text{sim}(Y^*, Y) \propto \frac{p(Y|Y^*)}{p(Y)}. \tag{10}$$

We want to put the representation function of mutual information into the NCE Loss

$$\mathcal{L}_N = -\mathbb{E}_{\mathbb{Y}} \log \left[ \frac{\text{sim}\,(Y^*, Y)}{\sum_{Y' \in \mathbb{Y}} (Y^*, Y')} \right],$$  (11)

where $\mathcal{L}_N$ denotes the NCE loss. By inserting the optimal $\text{sim}\,(Y^*, Y)$ into Eq. (11), we can get:

$$
\begin{aligned}
\mathcal{L}_{\text{NCE}} &= -\mathbb{E}_{\mathbb{Y}} \log \left[ \frac{\frac{p(Y|Y^*)}{p(Y)}}{\frac{p(Y|Y^*)}{p(Y)} + \sum_{Y' \in \mathbb{Y}_{\text{neg}}} \frac{p(Y^*,Y')}{p(Y')}} \right] \\
&= \mathbb{E}_{\mathbb{Y}} \log \left[ 1 + \frac{p(Y|Y^*)}{p(Y)} \sum_{Y' \in \mathbb{Y}_{\text{neg}}} \frac{p(Y^*,Y')}{p(Y')} \right] \\
&\approx \mathbb{E}_{\mathbb{Y}} \log \left[ 1 + \frac{p(Y|Y^*)}{p(Y)} (N-1) \mathbb{E}_{Y'} \frac{p(Y^*,Y')}{p(Y')} \right] \\
&= \mathbb{E}_{\mathbb{Y}} \log \left[ 1 + \frac{p(Y|Y^*)}{p(Y)} (N-1) \right] \\
&\geq \mathbb{E}_{\mathbb{Y}} \log \left[ \frac{p(Y|Y^*)}{p(Y)} N \right] \\
&= -I(Y^*, Y) + \log(N)
\end{aligned}
$$  (12)

$\square$

## C   Graph Mix-up Approach

To address the distribution shifting issue between $f(G)$ and $f(G^*)$ in the GIB objective, we introduce the mix-up approach to reconstruct a within-distribution graph, $G^{(\text{mix})}$, from the explanation graph $G^*$. We follow [24] to make a widely-accepted assumption that a graph can be divided by $G = G^* + G^\Delta$, where $G^*$ presents the underlying sub-graph that makes important contributions to GNN's predictions, which is the expected explanatory graph, and $G^\Delta$ consists of the remaining label-independent edges for predictions made by the GNN. Both $G^*$ and $G^\Delta$ influence the distribution of $G$. Therefore, we need a graph $G^{(\text{mix})}$ that contains both $G^*$ and $G^\Delta$, upon which we use the prediction of $G^{(\text{mix})}$ made by $f$ to approximate $Y^*$ and $\boldsymbol{h}^*$.

Specifically, for a target graph $G_a$ in the original graph set to be explained, we generate the explanation sub-graph $G_a^* = G_a - G_a^\Delta$ from the explainer. To generate a graph in the same distribution of original $G_a$, we can randomly sample a graph $G_b$ from the original set, generate the explanation sub-graph of $G_b^*$ with the same explainer and retrieve its label-irrelevant graph $G_b^\Delta = G_b - G_b^*$. Then we can merge $G_a^*$ together with $G_b^\Delta$ and produce the mix-up explanation $G_a^{(\text{mix})}$. Formally, we can have $G_a^{(\text{mix})} = G_a^* + (G_b - G_b^*)$.

Since we are using the edge weights mask to describe the explanation, we can denote $G_a$ and $G_b$ with the adjacency matrices $\boldsymbol{A}_a$ and $\boldsymbol{A}_b$, their edge weight mask matrices as $\boldsymbol{M}_a$ and $\boldsymbol{M}_b$. If $G_a$ and $G_b$ are aligned graphs with the same number of nodes, we can simply mix them up by $\boldsymbol{M}_a^{(\text{mix})} = \boldsymbol{M}_a^* + (\boldsymbol{I}_b - \boldsymbol{M}_b^*)$, where $\boldsymbol{M}$ denotes the weight of the adjacency matrix and $\boldsymbol{I}_b$ denotes the zero-ones matrix as weights of all edges in the adjacency matrix of $G_b$, where 1 represents the existing edge and 0 represents there is no edge between the node pair.

If $G_a$ and $G_b$ are not aligned with the same number of nodes, we can use a connection adjacency matrix $\boldsymbol{A}_{\text{conn}}$ and mask matrix $\boldsymbol{M}_{\text{conn}}$ to merge two graphs with different numbers of nodes. Specifically, the mix-up adjacency matrix can be formed as:

$$\boldsymbol{A}_a^{(\text{mix})} = \begin{bmatrix} \boldsymbol{A}_a & \boldsymbol{A}_{\text{conn}} \\ \boldsymbol{A}_{\text{conn}}^T & \boldsymbol{A}_b \end{bmatrix}.$$  (13)

And the mix-up mask matrix can be formed as:

$$\boldsymbol{M}_a^{(\text{mix})} = \begin{bmatrix} \boldsymbol{M}_a^* & \boldsymbol{M}_{\text{conn}} \\ \boldsymbol{M}_{\text{conn}}^T & \boldsymbol{M}_b^\Delta \end{bmatrix}$$  (14)

Finally, we can form $G_a^{\text{(mix)}}$ as $(\boldsymbol{X}^{\text{(mix)}}, \boldsymbol{A}_a^{\text{(mix)}} \odot \boldsymbol{M}_a^{\text{(mix)}})$, where $\boldsymbol{X}^{\text{(mix)}} = [\boldsymbol{X}_a; \boldsymbol{X}_b]$. The detailed algorithm for mix-up is shown in Algorithm 1. In implantation, $\eta = |\mathcal{E}| * 0.03$.

---

**Algorithm 1** Graph Mix-up Algorithm

---

**Input:** Target to-be-explained graph $G_a = (\boldsymbol{X}_a, \boldsymbol{A}_a)$, $G_b$ sampled from a set of graphs $\mathcal{G}$, the number of random connections $\eta$, explainer model $E$.
**Output:** Graph $G^{\text{(mix)}}$.
  1: Generate mask matrix $\boldsymbol{M}_a = E(G_a)$
  2: Generate mask matrix $\boldsymbol{M}_b = E(G_b)$
  3: Sample $\eta$ random connections between $G_a$ and $G_b$ as $\boldsymbol{A}_{\text{conn}}$
  4: Mix-up adjacency matrix $\boldsymbol{A}_a^{\text{(mix)}}$ with Eq. (13)
  5: Mix-up edge mask $\boldsymbol{M}_a^{\text{(mix)}}$ with Eq. (14)
  6: Mix-up node features $\boldsymbol{X}^{\text{(mix)}} = [\boldsymbol{X}_a; \boldsymbol{X}_b]$
  7: **return** $G^{\text{(mix)}} = (\boldsymbol{X}^{\text{(mix)}}, \boldsymbol{A}_a^{\text{(mix)}} \odot \boldsymbol{M}_a^{\text{(mix)}})$

---

# D  Training Algorithm

---

**Algorithm 2** Training Explainer

---

**Input:** A set of graphs $\mathcal{G}$, trained GNN model $f$, explainer model $E$.
**Output:** Trained explainer $E$.
  1: Initialize explainer model $E$.
  2: **for** $e \in epochs$ **do**
  3:     **for** $G \in \mathcal{G}$ **do**
  4:         $G_b, G_c \leftarrow$ Randomly sample two graphs from $\mathcal{G}$
  5:         $G^+, G^- \leftarrow$ Compare similarity$(G_b, G_c)$ to $G$
  6:         $G^{\text{(mix)}+} \leftarrow$ Mix-up $(G, G^+)$
  7:         $G^{\text{(mix)}-} \leftarrow$ Mix-up $(G, G^-)$
  8:         Compute $\mathcal{L}_{\text{NCE}}(G, G^+, G^-)$ with Eq. (6)
  9:         Compute $\mathcal{L}_{\text{GIB}}$ and overall loss $\mathcal{L}$ with Eq. (8)
10:     **end for**
11:     Update $E$ with back propagation.
12: **end for**
13: **return** Explainer $E$

---

Algorithm 2 shows the training procedure for our explainer. For each epoch and each to-be-explained graph $G$, we first randomly sample two neighbors and decide the positive neighbor $G^+$ and negative neighbor $G^-$ according to the similarity between their embedding vectors respectively. The graph with higher similarity to $G$ is the positive neighbor $G^+$. We generate the explanation for graphs and mix $G$ with $G^+$ and $G^-$ respectively. We calculate the InfoNCE for triplet $\langle G, G^+, G^- \rangle$ with Eq. (6) and the GIB loss, which contains the size loss and InfoNCE loss. We also calculate the MSE loss between $f(G^{\text{(mix)}+})$ and $f(G)$. The overall loss is the sum of size loss, InfoNCE loss, and MSE loss. We update the trainable parameters in the explainer with the overall loss.

# E  Implantation details

We provided implementation details for our experiments in this section. Data and code are available in the Anumuous git repo and supplementary.

All experiments are conducted on a Linux machine (Ubuntu 16.04.4 LTS (GNU/Linux 4.4.0-210-generic x86_64)) with 4 NVIDIA TITAN Xp (12 GB) GPUs. CUDA version is 11.8 and the Driver version is 520.56.06. All codes are written with the Python version 3.8.13 with PyTorch 1.12.1 and PyTorch Geometric (PyG) 2.1.0.post1, torch-scatter 2.0.9, and torch-sparse 0.6.15. We adopt the Adam optimizer throughout all experiments. Overall, for each dataset, a GCN regression model is well-trained first, where we take a **three-layer GCN** model as the backbone. Then the explainers take the to-be-explained GNN and original graph and generate explanations for its prediction on the

dataset. After that, we evaluate the performance of the explanation. We split the dataset into 8:1:1, where we train the GNN base model with 8 folds, and train and test explainer models with 1 fold respectively. The hyper-parameters are illustrated in the paper correspondingly.

## E.1 Datasets

We formulate Three synthetic datasets and a real-world dataset, as is shown in Table 1, in order to address the lack of graph regression datasets with ground-truth explanations. (1) *BA-Motif-Volume*: This dataset is based on the BA-shapes [23] and makes a modification, which is adding random float values from [0.00, 100.00] as the node feature. We then sum the node values on the motif as the regression label of the whole graph, which means the GNNs should recognize the [house] motif and then sum features to make the prediction. (2) *BA-Motif-Counting*: Different from BA-Motif-Volume, where node features are summarized, in this dataset, we attach various numbers of motifs to the base BA random graph and pad all graphs to equal size. The number of motifs is counted as the regression label. Additionally, we pad base graphs to dynamic size to prevent the GNNs from making trivial predictions based on the total number of nodes. (3) *Triangles*: We follow the previous work [39] to construct this dataset. The dataset is a set of 5000 Erdős–Rényi random graphs denoted as $ER(m, p)$, where $m = 30$ is the number of nodes in each graph and $p = 0.2$ is the probability for an edge to exist. The size of 5000 was chosen to match the previous work. The regression label for this dataset is the number of triangles in a graph and GNNs are trained to count the triangles. (4) *Crippen*: The Crippen dataset is a real-life dataset that was initially used to evaluate the graph regression task. The dataset has 1127 graphs reported in the Delaney solubility dataset [40] and has weights of each node assigned by the Crippen model [42], which is an empirical chemistry model predicting the water-actual partition coefficient. We adopt this dataset, firstly shown in the previous work [43], and construct edge weights by taking the average of the two connected nodes' weights.

## E.2 Baselines

We compared the proposed RegExplainer against a comprehensive set of baselines in all datasets, including: (1) **GRAD** [23]: GRAD is a gradient-based method that learns weight vectors of edges by computing gradients of the GNN's objective function. (2) **ATT** [41]: ATT is a graph attention network (GAT) that learns attention weights for edges in the input graph. These weights can be utilized as a proxy measure of edge importance. (3) **GNNExplainer** [23]: GNNExplainer is a model-agnostic method that learns an adjacency matrix mask by maximizing the mutual information between the predictions of the GNN and the distribution of possible sub-graph structures. (4) **PGExplainer** [24]: PGExplainer adopts a deep neural network to parameterize the generation process of explanations, which facilitates a comprehensive understanding of the predictions made by GNNs. It also produces sub-graph explanations with edge importance masks. (5) **MixupExplainer** [19]: MixupExplainer adopts the graph mix-up approach with PGExplainer and address the Out-Of-Distribution problem in graph classification tasks.

# F Additional Experiments

## F.1 Correlation between Prediction Shifting and the Label Value

In Figure 7, each point represents a graph instance, where $Y$ represents the ground-truth label, and $\Delta$ represents the absolute value difference. It's clear that both the $\Delta(f(G^*), Y)$ and $\Delta(f(G), f(G^*))$ strongly correlated to $Y$ with statistical significance, indicating the prediction shifting problem is related to the continuous ordered decision boundary, which is present in regression tasks.

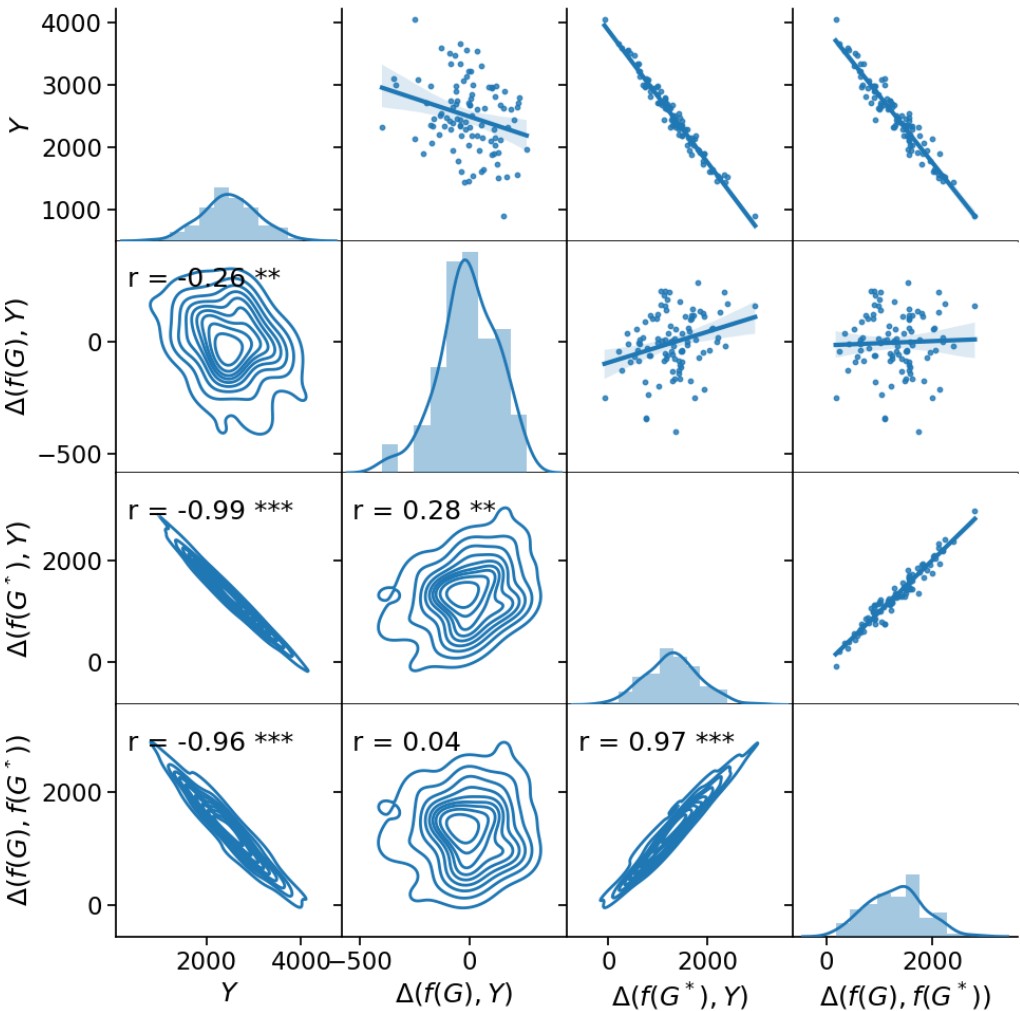

Figure 7: Correlations between the predictions and their shifting on BA-Motif-Volume. The value of $r$ indicates the Pearson Correlation Coefficient, and the values with * indicate statistical significance for correlation, where *** indicates the p-value for testing non-correlation $p \leq 0.001$. Each point represents one graph instance.

