# OpenReview forum: "RegExplainer: Generating Explanations for Graph Neural Networks in Regression Tasks"
_NeurIPS.cc/2024/Conference — NeurIPS 2024 poster_

### Official Review · Reviewer_4dqr · 2024-07-03

**Soundness:** 3
**Presentation:** 3
**Contribution:** 3
**Rating:** 5
**Confidence:** 2

**Summary:**

The authors focus on the interpretability of GNN in graph regression tasks. They propose a novel explanation method called RegExplainer as a plug-in to existing explanation methods, such as GNNExplainer and PGExplainer. They also tackle mutual information estimation (graph information bottleneck), distribution shifting and continuously ordered decision boundaries that hinder current explanation techniques.

**Strengths:**

1. The writing flow is clear and easy to follow.

2. The illustration in Figure 3 is clear and straightforward.

**Weaknesses:**

1. The **contribution** is confusing:

1a. In the introduction (Lines 64-68), the authors mention *datasets* as a key contribution. However, the paper lacks detailed descriptions of these datasets in later sections. More information about the datasets, including their structure and significance, should be included to clarify this contribution.

2. Concerning the **methods** and corresponding **experimental settings**:

2a. The *subgraph connectivities* are not considered. In other words, RegExplainer could result in isolated nodes rather than "a subgraph".

2b. **Property 2 is overclaimed**. Lines 469-471 dismiss the ground truth joint distribution $p(Y^*,Y)$, and only focus on the remaining parts. The assumption weakens the paper, especially since "distribution shifting" is a key discussion of the paper.

2c. Section 4.3, especially Step 2, is hard to follow. The process for obtaining $(G^+)^{\Delta}$ and $G^*$ needs more detail. Specifically, Step 2 is used to approximate $I(G, G^*)$ and find the optimal $G^*$ as in Eq. (4) or (5). However, $G^{(mix)}$ relies on $(G^+)^{\Delta}$ and $G^*$. Given that $G^*$ is the variable to be optimized, how to pick $(G^+)^{\Delta}$ is missing.

2d. Whether $G^*$, $(G^+)^{\Delta}$ ,as well as $G^{mix}$ are binary {0,1} or continuous in [0,1] is unclear. If they are binary, the optimization for $G^*$ needs more explanation. If they are continuous, the relaxation and recovery to the final outcomes should be specified.

2e. The method of measuring distances between graphs or graph distributions is not clearly defined. The authors should clarify and justify whether MSE/RMSE are used, or another metric is needed.

Besides, the authors evaluate the distribution shifting by similarity/distance between embeddings (Table 3) rather than other distribution-based metrics.

2f. The last column in Table 2 shows a large distance from the original prediction, suggesting the *subgraph may not be faithful*.

2g. The problem definition in Lines 113-115 is unclear, how to define "can" explain?
The problem definition in Lines 113-115 is unclear, particularly the criteria for an explanation to be considered valid (e.g., “can explain the prediction”).

3. **Others**:

3a. The method appears suited for graph-level regression tasks. Could it apply to the node-level regression tasks, such as traffic prediction?

3b. "N" in Line 110 is undefined.

3c. What does "regression label in regression task" mean on Line 164?

**Questions:**

Please address the concerns in the "Weakness".

**Limitations:**

The authors show the limitations on Sec. 6.

However, as mentioned above, the assumption and proof of Property 2 should be carefully considered.

---

> ### Author Rebuttal · Authors · 2024-08-07
>
> Dear reviewer 4dqr, thank you for taking the time to review our work and providing feedback. In the following, we aim to address your questions and concerns.
>
> A-1a: Thank you for your suggestion. We will include detailed descriptions of the datasets in future versions of the paper. In the appendix, we describe how these datasets were created and the rationale behind their creation.
>
> A-2a: Thank you for pointing this out. Subgraph connectivity is indeed an important issue. However, in this work, our main goal was to extend GIB-based explainers to graph regression tasks, following the settings of previous works in other aspects. We are keen to explore and address the connectivity of explanatory subgraphs in future research.
>
> A-2b: We incorporate the InfoNCE loss into the GIB objective with reference to previous work[1], where InfoNCE is a lower bound of mutual information (MI), and there is no constraint on the distribution of variables. Thank you for pointing out the issue in the appendix. We will provide a more detailed explanation in the next version.
>
> A-2c: We are happy to address your concern. In Step 1 of Section 4.3, we describe in detail how to select $G^+$ and $G^-$. “We can define two randomly sampled graphs as positive neighbor $G^+$ and negative neighbor $G^−$, where $G^+$’s label $Y^+$ is closer to $Y$ than $G^−$’s label $Y^−$, i.e., $|Y^+ − Y| < |Y^− − Y|$.”
>
> A-2d: They are continuous. The final explanatory subgraph is obtained by selecting the top-k edges based on their weights.
>
> A-2e: Thank you for pointing this out. We will correct the typo in Table 3 in the next version of the paper. We use RMSE as the metric for measuring distances between graphs or graph distributions. Besides, metrics like KL-divergence can be challenging to compute in practice. Therefore, we use graph embeddings and prediction labels to measure distribution shifts between original graphs and explanation subgraphs.
>
> A-2f: The purpose of Table 2 is to demonstrate that the explanatory subgraphs can indeed be out-of-distribution (OOD). The results in the last column indicate that these subgraphs are not sufficiently faithful and can introduce bias in the explaining. To address this issue, we introduced the mixup method to mitigate the OOD problem of the explanatory subgraphs.
>
> A-2g: "Can explain" means that the explanation should identify the reasons behind the GNN's prediction. For example, in the defined datasets, different motifs and their corresponding features lead to different labels. The GNN relies on these motifs to make predictions, and the goal of the explainer is to identify these motifs as part of the explanation.
>
> A-3a: Yes, our method can be extended to node-level regression tasks. In the case of a three-layer GCN, node-level tasks essentially involve graph tasks centered on 3-hop subgraphs around each node. Therefore, our method can be easily adapted for node-level tasks. We look forward to incorporating more datasets in future work.
>
> A-3b: Thank you for pointing this out. "N" represents the number of graphs in the dataset. We will include this clarification in the next version of the paper.
>
> A-3c: The term "regression label" refers to the prediction made by the GNN for a given graph or subgraph.
>
> [1] Aaron van den Oord, Yazhe Li, and Oriol Vinyals. 2018. Representation Learning with Contrastive
> Predictive Coding.

---

> > ### Comment · Reviewer_4dqr · 2024-08-08
> >
> > Thanks for the responses. However, some of the answers do not fully address my concerns.
> >
> > > To A-2b:
> >
> > If it cannot handle, or ignore, the joint distribution, how could you justify the generality of Property 2?
> >
> > > To A-2c:
> >
> > Only the processes of picking $G^+$ and $G^-$ are included, while the processes of picking $(G^{+})^\Delta$ is missing.
> >
> > > To A-2d:
> >
> > Such statements are missing in the manuscript.
> >
> > > To A-2e:
> >
> > The metrics should be further justified, since it is the key for evaluating the graph distributions, as well as the methods.
> > For example, is it possible to quickly survey the metrics used in the papers for the same tasks?
> >
> > > To A-2g:
> >
> > Is it possible to give a more formal definition or reference? E.g. if the task desires a lower/higher value under some specific evaluation measurements (please make it more clear and formal).
> >
> > > To A-3c:
> >
> > It is interesting to refer "regression *label*" to the *prediction* of a GNN.

---

> > > ### Author Response · Authors · 2024-08-13
> > > **Second Round Reply (1/2)**
> > >
> > > Dear reviewer 4dqr,
> > >
> > > We are glad to have further discussion with you. Your insightful comments are valuable to us and help improve the quality of our paper. We are grateful for the time and effort you have invested in providing such a thorough review. If you have any remaining concerns or require further clarification, please let us know. We are more than happy to address any additional questions you may have.
> > >
> > > > Re to A-2b:
> > >
> > > A: We see your concern regarding property 2. In this property, to develop the GIB objective into graph regression tasks, we firstly have $I(Y^*; Y) = \sum_{Y^*,Y} p(Y^*,Y) \log \frac{p(Y|Y^*)}{P(Y)}$, then, follow the previous work, we apply the “proportional to” trick [1] because $p(Y^*,Y)$ is an unbounded value. We then proportionally optimize $ \sum_{Y^*,Y} p(Y^*,Y) \log \frac{p(Y|Y^*)}{P(Y)}$ with $\text{sim}\left(Y^*, Y\right) \propto \frac{p(Y|Y^*)}{p(Y)}$. This objective is specifically designed for graph regression tasks, and we plan to include more synthetic and real-world datasets in our study to provide a more comprehensive evaluation. Currently, the performance on three synthetic datasets and one real-world dataset demonstrates its effectiveness.
> > >
> > >
> > >
> > > > Re to A-2c:
> > >
> > > A: Thank you for your further question. For graph sample G+, we have $(G^+)^*=E(G+)$, where $(G^+)^*$ is the label-preseving subgraph of $G^+$. Then we pick the $(G^+)^{\Delta}=G^+ - (G^+)^*$ as the label-irrelevant subgraph. This procedure is included in the equations above line189. Eg: $G^{\text{(mix)}+}=G^*+ (G^+)^{\Delta} = G^*+(G^+-(G^+)^*)$.
> > > We will provide a more detailed description of this process in Section 4.3, Step 1, in the next version of our paper.
> > >
> > > > Re to A-2d:
> > >
> > > A: Thank you for pointing this out. We mentioned that we are following previous work and briefly described this from lines 119 to 122 in the paper. We will ensure these clarifications are more explicitly detailed in the next version of the paper.
> > >
> > >
> > > > Re to A-2e:
> > >
> > > A: Thank you for your insightful suggestion. In response, we conducted a survey of metrics commonly used in similar tasks within the literature. The most prevalent metrics include:
> > > Graph Edit Distance (GED)
> > > Wasserstein Distance (Earth Mover's Distance)
> > > Maximum Mean Discrepancy (MMD)
> > > Jensen-Shannon Divergence (JSD, a variant of KL divergence)
> > > Graph Kernel Methods
> > > These metrics vary in complexity and applicability depending on the nature of the graphs and the specific task at hand. Given the context of our work, we believe that MMD might be particularly suitable, while Graph Kernel Methods could also provide valuable insights into graph structure shifts.
> > >
> > > **We are currently conducting experiments to evaluate the MMD metric, and we will include these results in the supplementary material of the next version of the paper.** This will help us better justify our choice of metrics and possibly introduce additional metrics for a more comprehensive evaluation.
> > >
> > > We appreciate your recommendation and will incorporate these findings into the revised manuscript to enhance the clarity and rigor of our evaluation process.
> > >
> > >
> > > > Re to A-2g:
> > >
> > > A: We adhere to established definitions from previous work when evaluating the explainability of GNNs. For datasets with explanation ground truth, we use the *AUC-ROC* (Area Under the Receiver Operating Characteristic Curve) as the evaluation metric [2, 3] to measure explanation performance, where a higher AUC-ROC score indicates better performance.
> > >
> > > For datasets without ground truth explanations, we utilize metrics such as the fidelity-sparsity score and the robust-fidelity score [4] to assess the quality of the explanations. These metrics are designed to balance the trade-off between explanation fidelity (how well the explanation reflects the model's predictions) and sparsity (how concise the explanation is).
> > >
> > > In our work, the AUC-ROC metric is employed, with higher scores reflecting superior explanation performance. We rely on this metric because it has been widely accepted in the literature as a standard measure for evaluating the quality of model explanations.

---

> > > ### Author Response · Authors · 2024-08-13
> > > **Second Round Reply (2/2)**
> > >
> > > > Re to A-3c:
> > >
> > > A: This term is also used in previous works[5, 6, 7, 8]. We will make a clear description in the next version of the paper.
> > >
> > > [1]. Aaron van den Oord, Yazhe Li, and Oriol Vinyals. 2018. Representation Learning with Contrastive Predictive Coding.
> > >
> > > [2]. Zhitao Ying, Dylan Bourgeois, Jiaxuan You, Marinka Zitnik, and Jure Leskovec. Gnnexplainer: Generating explanations for graph neural networks. Advances in neural information processing systems, 32, 2019
> > >
> > > [3]. Dongsheng Luo, Wei Cheng, Dongkuan Xu, Wenchao Yu, Bo Zong, Haifeng Chen, and Xiang Zhang. Parameterized explainer for graph neural network. Advances in neural information processing systems, 33:19620–19631, 2020.
> > >
> > > [4]. Xu Zheng, Farhad Shirani, Tianchun Wang, Wei Cheng, Zhuomin Chen, Haifeng Chen, Hua Wei, Dongsheng Luo. Towards Robust Fidelity for Evaluating Explainability of Graph Neural Networks.
> > >
> > > [5]. Haoliang Yuan, Junjie Zheng, Loi Lei Lai, Yuan Yan Tang. A constrained least squares regression model.
> > >
> > > [6]. Cheng Li, Virgil Pavlu, Javed Aslam, Bingyu Wang & Kechen Qin. Learning to Calibrate and Rerank Multi-label Predictions.
> > >
> > > [7]. Xin Ding, Yongwei Wang, Zuheng Xu, William J. Welch, Z. Jane Wang. Continuous Conditional Generative Adversarial Networks: Novel Empirical Losses and Label Input Mechanisms.
> > >
> > > [8]. Harikrishna Narasimhan, Andrew Cotter, Maya Gupta, Serena Wang. Pairwise Fairness for Ranking and Regression.

---

> > > > ### Comment · Reviewer_4dqr · 2024-08-13
> > > >
> > > > Thank you for the clarification! I will raise my score.

---

### Official Review · Reviewer_X7ek · 2024-07-09

**Soundness:** 4
**Presentation:** 3
**Contribution:** 3
**Rating:** 6
**Confidence:** 4

**Summary:**

This work proposes a method to generate instance-level GNN prediction explanations specifically for graph regression tasks. This method addresses distribution shifting, a problem in regression, by using mix-up for contrastive learning. The work is evaluated on four datasets, both synthetic and real-world.

**Strengths:**

The work provides thorough theoretical justification for the objective function in Eq. (4). Additionally, it creates a clever mix-up-based contrastive learning approach that directly addresses the distribution shift problem specific to regression.

**Weaknesses:**

More real-world datasets would demonstrate this method’s more general applicability. The evaluation is limited by the inclusion of only one real-world dataset, Crippen.

See questions.

**Questions:**

How does this method perform with/against classification explanation models? As the explainer can be modularly applied to existing trained GNNs, this would be interesting to see if it can improve the accuracy of explanation generations. This work [1] is a good work on instance and model-level explanations for graph classification tasks.

Is graph mix-up performed on the graphs explicitly or within latent space? If the mix-up occurs in graph space, then how is the subgraph G* combined with (G^+)^{Delta} and (G^-)^{Delta}? How are edges added between these different subgraphs? Furthermore, for more complex regression datasets, naively adding edges can drastically change the label for each graph. Is this strategy then limited to graph regression datasets which inherently rely on graph structures to derive labels? If the mix-up occurs within latent space, then how does this work differentiate itself from graph rationalization works [2, 3]? If the mix-up occurs in latent space, then additional studies to compare against graph rationalization baselines should be included.

[1] Xuanyuan, Han, et al. "Global concept-based interpretability for graph neural networks via neuron analysis." Proceedings of the AAAI Conference on Artificial Intelligence. Vol. 37. No. 9. 2023.

[2] Wu, Ying-Xin, et al. "Discovering invariant rationales for graph neural networks." arXiv preprint arXiv:2201.12872 (2022).

[3] Liu, Gang, et al. "Graph rationalization with environment-based augmentations." Proceedings of the 28th ACM SIGKDD Conference on Knowledge Discovery and Data Mining. 2022.

**Limitations:**

The work requires an accurate trained prediction model. The model explanations are not used to retrain the GNN in any way.

Additional limitations are addressed in the conclusion.

---

> ### Author Rebuttal · Authors · 2024-08-07
>
> Dear reviewer X7ek, thank you for taking the time to review our work and providing feedback. In the following, we aim to address your questions and concerns.
>
> > How does this method perform with/against classification explanation models? As the explainer can be modularly applied to existing trained GNNs, this would be interesting to see if it can improve the accuracy of explanation generations. This work [1] is a good work on instance and model-level explanations for graph classification tasks.
>
>
> A1: Yes, we have compared our method with several existing methods for graph classification, including GRAD, ATT, MixupExplainer, GNNExplainer, and PGExplainer, as shown in Table X. It is important to note that these methods were originally designed for graph classification tasks and use cross-entropy loss in their objective functions, which cannot be directly applied to graph regression tasks. Therefore, we adapted them for comparison by using MSE loss.
>
> Work [1] is an excellent study that explores graph neural network interpretability from a novel perspective. Our method could potentially be adapted to the framework presented in their Equation 6 and might improve the performance of their approach. We look forward to incorporating and citing this method as a baseline in our future work.
>
> > Is graph mix-up performed on the graphs explicitly or within latent space? If the mix-up occurs in graph space, then how is the subgraph G* combined with $(G^+)^{\Delta}$ and $(G^-)^{\Delta}$? How are edges added between these different subgraphs?
>
> A2: We describe the mix-up process in detail in Appendix C. The mix-up is performed on the graph explicitly by mixing edge weights, and different subgraphs are connected through randomly sampled connection edges.
>
> > Furthermore, for more complex regression datasets, naively adding edges can drastically change the label for each graph. Is this strategy then limited to graph regression datasets which inherently rely on graph structures to derive labels?
>
> A3: Yes, in this work, we follow the settings of previous studies and focus primarily on the graph structure without incorporating edge features. We are interested in exploring the impact of edge features on our method's performance in future research.
>
> > If the mix-up occurs within latent space, then how does this work differentiate itself from graph rationalization works [2, 3]? If the mix-up occurs in latent space, then additional studies to compare against graph rationalization baselines should be included.
>
> A4: Our mix-up primarily occurs on edge weights. We will consider more related works[2, 3] and incorporate their strengths and citations in future studies.
>
> >Limitations: The work requires an accurate trained prediction model. The model explanations are not used to retrain the GNN in any way.
>
> A5: Thank you for your suggestion. This work primarily focuses on post-hoc explanations. In future work, we will explore methods that involve retraining the GNN to enhance both the GNN and the explainer's performance.

---

> > ### Comment · Reviewer_X7ek · 2024-08-12
> >
> > Thank you to the authors for their rebuttal. After reading all of the reviews and responses, I will choose to maintain my score.

---

> > > ### Author Response · Authors · 2024-08-13
> > >
> > > Dear reviewer X7ek,
> > >
> > > Thank you very much for your reply. We truly appreciate the time and effort you invested in evaluating our work and providing valuable feedback.
> > >
> > > Although the score did not change, your constructive comments are invaluable to us, and we are committed to addressing all your concerns thoroughly. We are grateful for your support and the opportunity to enhance our work.
> > >
> > > If you have any further suggestions or need additional clarifications, please let us know. We are more than happy to provide any additional information or address any remaining questions.

---

### Official Review · Reviewer_B4J7 · 2024-07-11

**Soundness:** 2
**Presentation:** 2
**Contribution:** 2
**Rating:** 5
**Confidence:** 2

**Summary:**

The authors propose an explanation method to interpret the graph regression models. The techniques are built upon the information bottleneck theory and contrastive learning. The authors show that their explanations are accurate in five graph regression datasets.

Note: If authors address my concerns in questions and limitations, I am willing to upgrade my allocated score.

**Strengths:**

Explanation of graph neural networks and, particularly, explaining graph regression is both a relevant and interesting problem.

The major parts of the proposed technique are clearly explained and easily understood in detail.

The proposed method relies on some solid theoretical properties like information bottleneck.

The code is released with the paper, which helps the work to be reproducible.

**Weaknesses:**

The paper is densely written in Section 4 and is hard to follow. Instead of explaining each part of the algorithm separately and clearly explaining why these components exist, the authors rely too much on theorizing the problem. For example, I am unsure if a reader is particularly interested in knowing about lower and upper bounds and distribution shifts before seeing the proposed method, and maybe moving this section away as motivation can solve this problem.

The authors have mentioned that their evaluation relies on ground truth, but the measures for evaluating explanations are barely discussed. See questions below.

**Questions:**

Where are the ground truth vectors for your evaluation? How are they obtained? How can ground truth for explanations be in the datasets? This is the most important part of your paper, and it is left to the imagination of readers.

I argue that your technique is better because you add more samples, and the surrogate is learning just that. So basically, you can add this sampling to those explanations, and you don't need the rest of your method (evidence in Section 5.3 Figure 5 except for the BA-motiv dataset). Can you argue against this concern?

Why are the ablation results different for BA-Motiv-Volume?

What is the real limitation of your technique? You have written, "Instead, they write, " Specifically, although our approach can be applied to explainers for graph regression tasks in an explainer-agnostic manner, it cannot be easily applied to explainers built for explaining the spatio-temporal graph due to the dynamic topology and node features of the STG." What does this mean?

**Limitations:**

I personally think the main limitation is that this approach is an add-on approach. I am not in favor of add-on model agnostic approaches. What I mean is explanations that sit on top of other explanation techniques. I think this makes the design of these techniques extra complicated, and finding faults in explanations becomes harder. I think if the authors take their sampling and improve it, it can replace the GNN and PGE and become a more general approach.

The authors have also not stated that the graph regression task is not a very popular task in graph settings. Based on this, I also would like to see how this approach can be extended for use in other graph tasks: node classification etc.

---

> ### Author Rebuttal · Authors · 2024-08-07
>
> Dear reviewer B4J7, thank you for taking the time to review our work and providing feedback. We appreciate your thorough review and aim to address your questions and concerns in detail. Due to character limitations, we will provide a detailed response to the remaining points in the official comment section. Please let us know if you have any further questions or need additional clarification, we are happy to discuss with you.
>
> > The paper is densely written ... problem.
>
> A1: Thank you very much for your suggestion. In this work, our goal is to provide reliable explanations for graph regression tasks.  Previous methods [1, 2, 3] used the vanilla GIB for graph classification, which cannot be trivially applied to graph regression tasks. Therefore, we introduced InfoNCE loss into GIB and theoretically validated its effectiveness. Next, we explain that the OOD problem is more severe in graph regression tasks. On this basis, we introduce the mixup method and combine it with InfoNCE loss to propose our method and model, which includes a contrastive learning objective function with contrastive loss. We aim to balance the effectiveness of the model design with the reliability of the theoretical foundation. In future versions, we will improve the organization of our paper, provide more detailed explanations of the model design, and move some of the theoretical derivations to the appendix.
>
> > Where are the ground truth vectors for your evaluation? How are they obtained? How can ground truth for explanations be in the datasets? This is the most important part of your paper, and it is left to the imagination of readers.
>
> A2: (1). What the explanation subgraphs look like: As introduced in Section 3 Preliminary, line 119 - line 120, we follow the setup from previous work[1, 2], using a binary edge mask to represent the explanation subgraph of the original graph. Specifically, for each edge in the original graph, our explainer produces a prediction value. If this value is 1, it indicates that the edge is part of the explanation; if it is 0, it means the edge is not relevant to the explanation. For each graph, the mask containing these edge weight values is a vector.
>
> (2)&(3). How the ground truth for explanation subgraphs is obtained: Establishing the ground truth for explanation subgraphs is a critical step. We adopt the approach from previous work, including both synthetic and real datasets. In synthetic datasets like BA-motif-volume, we designed the dataset such that the label is related to the motifs and their corresponding features within the graph. Therefore, the corresponding motif subgraph is the explanation subgraph and is used to evaluate the explainer's performance. Additionally, we designed a graph regression task with ground truth explanation subgraphs based on the chemical dataset Crippen. More detailed information can be found in Appendix E.1.
>
> (4). How we evaluate the explainer’s performance and what metrics are used: As mentioned earlier, the explainer produces a weight for each edge in the graph and determines whether the edge belongs to the explanation subgraph based on this weight. In practice, the edge weights are floating-point numbers, and we assess the accuracy of the explanation by calculating the AUC-ROC with respect to the ground truth, thus evaluating the explainer’s performance.
>
> (5). Additional Information: In some other works, evaluation methods without ground truth are used, such as fidelity/sparsity scores to estimate the quality of the explanation subgraph. However, these evaluation methods face issues with out-of-distribution (OOD) problems. Therefore, we did not use datasets and metrics without ground truth in this work. We plan to introduce more datasets, both with and without ground truth, in future work to better evaluate our method and facilitate related research.
>
> > I argue ... concern?
>
> A3: From my understanding, your concern means that you believe the improvement in explainer performance comes from sampling and mixup, and thus the contrastive learning and contrastive loss components are unnecessary. Let me clarify this point; if my understanding is incorrect, I would be happy to further discuss and clarify the issue with you.
>
> First, as mentioned in A1, our method aims to explain graph regression tasks, whereas previous methods were based on vanilla GIB and focused on explaining graph classification tasks. By introducing InfoNCE loss, we can better explain graph regression tasks. The sampling and mixup are employed to address the OOD (out-of-distribution) problem during the explanation process. The combination of these components is crucial for effectively improving the model's performance.
>
> As shown in Figure 5 of the ablation study, simply using the mixup method and sampling (dropping the InfoNCE module) leads to a decrease in model performance. Therefore, just sampling is not sufficient to fully address the problem.
>
> [1] Zhitao Ying, Dylan Bourgeois, Jiaxuan You, Marinka Zitnik, and Jure Leskovec. Gnnexplainer: Generating explanations for graph neural networks. Advances in neural information processing systems, 32, 2019.
>
> [2] Dongsheng Luo, Wei Cheng, Dongkuan Xu, Wenchao Yu, Bo Zong, Haifeng Chen, and Xiang Zhang. Parameterized explainer for graph neural network. Advances in neural information processing systems, 33:19620–19631, 2020.
>
> [3] Hao Yuan, Haiyang Yu, Jie Wang, Kang Li, and Shuiwang Ji. On explainability of graph neural networks via subgraph explorations. In International Conference on Machine Learning, pages 12241–12252. PMLR, 2021.

---

> > ### Comment · Reviewer_B4J7 · 2024-08-11
> >
> > Thank you for your response. You have addressed some of my concerns, and I can raise my score to +1—best of luck.

---

> > > ### Author Response · Authors · 2024-08-13
> > >
> > > Dear reviewer B4J7,
> > >
> > > Thank you very much for your thoughtful review and for increasing the score of our paper. We sincerely appreciate the time and effort you invested in evaluating our work and providing valuable feedback. Your positive assessment and constructive comments have been instrumental in helping us improve the paper.

---

> ### Author Response · Authors · 2024-08-07
> **Rebuttal Part 2**
>
> > Why are the ablation results different for BA-Motiv-Volume?
>
> A4: This is a very good observation, and I am glad you pointed it out so we can explain it. We will include the relevant explanation in future versions to improve the quality of the paper.
>
> First, we observed that in the BA-motif-volume dataset and the Crippen dataset, the mixup module has a greater impact on method performance. In contrast, in the other two datasets, the InfoNCE loss (contrastive learning) has a more significant influence. We believe this is due to the characteristics of the datasets: in BA-motif-volume, all graphs are of the same size. Therefore, for a well-trained GNN, the explanation subgraphs (which are only parts of the original graphs) are significantly out-of-distribution (OOD), which reduces model and explainer performance. By using the mixup method to address this issue, we can effectively improve performance. In the other two datasets, the graph sizes are dynamic, and the trained GNNs are relatively more robust, so the performance loss caused by not using mixup is not as substantial.
>
> > What is the real ... mean?
>
> A5: This means that our method cannot be trivially applied to tasks involving dynamic graph structures, such as spatio-temporal graphs (STGs). In STGs, the graph structure changes over time, which poses challenges for selecting samples for contrastive learning and mixup. The dynamic nature of the topology and node features makes it difficult to apply our approach directly. We aim to extend our method to handle dynamic graph structures in future work.
>
> > Limitations
>
> A6: I understand your concern. However, the add-on approach actually offers advantages. Unlike single and fixed explainer methods, our framework can be flexibly applied to existing explainer models, making it suitable for graph regression tasks and improving their performance. In our code, we have also included implementations of Regexplainer based on GNNExplainer and PGExplainer. These implementations can directly replace the original GNNExplainer and PGExplainer and be used straightforwardly.
>
> > The authors have also not stated ... etc.
>
> A7: Our method is specifically designed to improve explaining the graph regression tasks. For graph classification tasks, we can adapt our approach by replacing the MSE loss in the objective function with cross-entropy loss. Additionally, our method can be extended to node-level tasks. For example, in a 3-layer GNN, a node classification task can be viewed as a graph task on 3-hop subgraphs centered around nodes, and then a variant of our method with cross-entropy loss could be easily adapted to the dataset.

---

### Official Review · Reviewer_LTFm · 2024-07-13

**Soundness:** 3
**Presentation:** 3
**Contribution:** 3
**Rating:** 5
**Confidence:** 2

**Summary:**

The paper introduces XAIG-R, a novel explanation method for interpreting graph regression models. It addresses distribution shifting and decision boundary issues, leveraging the graph information bottleneck theory (GIB) and self-supervised learning.

**Strengths:**

- Intuitive and clear presentation and illstrration.
- Previous works have primarily focused on explaining GNN models in classification tasks, leaving a gap in understanding graph regression models. This paper specifically targets the explanation of graph regression tasks, addressing a previously unexplored area.
- Extensive Experimental Validation with well-designed settings.

**Weaknesses:**

- Limited Discussion on Computational Efficiency

**Questions:**

- In this work, GIB is used as Explainer, have you also studied other instance-based approach like GNNExplainer etc.
-  How does the proposed method handle dynamic graph topology changes, and what are the implications for real-world applications with evolving graph structures? (Directed Graph, Hyper Graph)
- Any other general-used datasets are tested? Especially those real-world graph datasets.

**Limitations:**

The authors adequately addressed the limitations.

---

> ### Author Rebuttal · Authors · 2024-08-07
>
> Dear reviewer LTFm, thank you for taking the time to review our work and providing feedback. In the following, we aim to address your questions and concerns.
>
> > Limited Discussion on Computational Efficiency
>
> A1: Thank you for pointing this out. We are glad to supplement our analysis of computational complexity, which will also be included in future versions.
> In the implementation, we transform the structure of the graph data from the sparse adjacency matrix representation into the dense edges list representation. We analyze the computational complexity of our mix-up approach here.
>
> Given a graph $G_a$ and a randomly sampled graph $G_b$, assuming $G_a$ contains $M_a$ edges and $G_b$ contains $M_b$ edges, the complexity of graph extension operation on edge indices and masks, which extend the size of them from $M_a$, $M_b$ to $M_a+M_b$, is $O(2(M_a+M_b))$, where $M_a>0$ and $M_b>0$. To generate $\eta$ cross-graph edges, the computational complexity is $O(\eta)$. For the mix-up operation, the complexity is $O(2(M_a+M_b)+\eta)$.
> Since $\eta$ is usually a small constant, the time complexity of our mix-up approach is $O(2*M_a+2*M_b)$.
>
> We use $M$ to denote the largest number of edges for the graph in the dataset and the time complexity of mix-up could be simplified to $O(M)$.
>
> > In this work, GIB is used as Explainer, have you also studied other instance-based approach like GNNExplainer etc.
>
> A2: GIB is a widely used theoretical foundation in related work. Both GNNExplainer and PGExplainer are based on GIB. In our experiments, we considered both GNNExplainer and PGExplainer and included a comparative study applying our method to GNNExplainer. The results are presented in Table 1 in the paper, as rows "GNNExplainer", "PGExplainer" and "+RegExplainer".
>
> > How does the proposed method handle dynamic graph topology changes, and what are the implications for real-world applications with evolving graph structures? (Directed Graph, Hyper Graph)
>
> A3: Thank you for pointing this out. We also discuss the issue of dynamic graphs in the limitations section. This is a very valuable problem, and we plan to further investigate the explainability of dynamic graph topology, evolving graph structures, and spatio-temporal graphs. This is our direction for future work.
>
> > Any other general-used datasets are tested? Especially those real-world graph datasets.
>
> A4: We would very much like to include more real-world datasets. However, the fact is that graph regression datasets containing ground truth explanation subgraphs are very rare. In this work, we created Crippen[1] as a real-world dataset. We will strive to discover, generate, and use more real-world datasets in future work.
>
> [1] John S Delaney. Esol: estimating aqueous solubility directly from molecular structure. Journal of chemical information and computer sciences, 44(3):1000–1005, 2004.

---

> > ### Author Response · Authors · 2024-08-13
> >
> > Dear Reviewer LTFm,
> >
> > Thank you once again for your detailed and insightful feedback. We are committed to addressing all concerns and ensuring the highest quality of our work. Your comments have been incredibly valuable, and we have made clarifications and provided more analysis based on your suggestions.
> >
> > To ensure we fully meet your expectations, could you please provide any further feedback or confirm if the revisions address your concerns? Your prompt response would be greatly appreciated as we finalize our revisions.
> >
> > Thank you for your time and effort.

---

### Official Review · Reviewer_KBtu · 2024-07-19

**Soundness:** 2
**Presentation:** 3
**Contribution:** 2
**Rating:** 5
**Confidence:** 3

**Summary:**

The paper addresses the challenge of interpreting graph regression models, a fundamental yet less explored task in graph learning compared to classification tasks. Existing explanation techniques are predominantly designed for classification, resulting in a gap for regression tasks. Based on the recent advances on information bottleneck and mix-up framework, the author proposes a novel objective to interprete GNN models in regression tasks.

**Strengths:**

1. The Performance of RegExplainer is exceptionally good compared with existing baselines in Table 1, which supports the claim of the paper well.

2. The background knowledge and related work summarization is comprehensive and easy to follow.

**Weaknesses:**

1. Some of the model designs are not moviated consistently over a pronounced challenge. The paper seems to be a combination of Mixupexplainer, GNNExplainer and G-Mixup.

2. The challgenges of graph regression explainer and distribution shift seem to be independent. The author doesn't make any justification on the graph topology and only embedding/predicted values are reported in Figure 3.

**Questions:**

1. In Figure 6, why the selection of $\alpha$ seems do not affect the overall performance? It seems the InfoNCE loss is negeligile in the optimization of the proposed method

---

> ### Author Rebuttal · Authors · 2024-08-07
>
> Dear reviewer KBtu, thank you for taking the time to review our work and providing feedback. In the following, we aim to address your questions and concerns.
>
> > Some of the model designs are not moviated consistently over a pronounced challenge. The paper seems to be a combination of Mixupexplainer, GNNExplainer and G-Mixup.
>
> A1: In this work, our goal is to provide reliable explanations for graph regression tasks. Previous methods [1, 2, 3], particularly the vanilla GIB used for graph classification, cannot be trivially applied to graph regression tasks. Therefore, we introduced InfoNCE loss into GIB and theoretically validated its effectiveness. We then recognized that the OOD (out-of-distribution) problem is more severe in graph regression tasks, so we incorporated the mixup method and combined it with InfoNCE loss to develop a contrastive learning objective function that includes contrastive loss. Our model design is cohesive and aims to address a specific problem, rather than simply combining Mixupexplainer, GNNExplainer, and G-Mixup.
>
> > The challgenges of graph regression explainer and distribution shift seem to be independent. The author doesn't make any justification on the graph topology and only embedding/predicted values are reported in Figure 3.
>
> A2: The explainability of graph regression and distribution shift are not independent issues. We found that the distribution shift problem is more severe in graph regression tasks. To generate better explanation subgraphs, we need to correct the subgraph distribution during the explanation process to avoid biased interpretations. Figure 3 illustrates why addressing the distribution shift problem is necessary. It shows that the GNN makes completely incorrect predictions for the BA-Motif-Volume samples, which significantly affects our estimation of $I(Y^*, Y)$ and leads the objective function in the wrong direction. Therefore, we need to address the explainability of graph regression and the distribution shift of subgraphs together.
>
> > In Figure 6, why the selection of \alpha seems do not affect the overall performance? It seems the InfoNCE loss is negeligile in the optimization of the proposed method
>
> A3: This is because our method exhibits strong robustness to the selection of hyperparameters. However, this does not imply that the InfoNCE loss is negligible. As shown in the ablation study in Figure 5, the performance of the method significantly decreases when the InfoNCE loss module is removed. Therefore, Figures 5 and 6 together demonstrate the robustness and effectiveness of the InfoNCE loss.
>
> [1] Zhitao Ying, Dylan Bourgeois, Jiaxuan You, Marinka Zitnik, and Jure Leskovec. Gnnexplainer: Generating explanations for graph neural networks. Advances in neural information processing systems, 32, 2019.
>
> [2] Dongsheng Luo, Wei Cheng, Dongkuan Xu, Wenchao Yu, Bo Zong, Haifeng Chen, and Xiang Zhang. Parameterized explainer for graph neural network. Advances in neural information processing systems, 33:19620–19631, 2020.
>
> [3] Hao Yuan, Haiyang Yu, Jie Wang, Kang Li, and Shuiwang Ji. On explainability of graph neural networks via subgraph explorations. In International Conference on Machine Learning, pages 12241–12252. PMLR, 2021.

---

> > ### Author Response · Authors · 2024-08-13
> >
> > Dear Reviewer KBtu,
> >
> > Thank you once again for your detailed and insightful feedback. We are committed to addressing all concerns and ensuring the highest quality of our work. Your comments have been incredibly valuable, and we have made clarifications based on your suggestions.
> >
> > To ensure we fully meet your expectations, could you please provide any further feedback or confirm if the revisions address your concerns? Your prompt response would be greatly appreciated as we finalize our revisions.
> >
> > Thank you for your time and effort.

---

> ### Author Response · Authors · 2024-08-13
> **Respectfully Requesting an Update**
>
> Dear reviewer KBtu,
>
> Thank you for your valuable feedback on our paper. We truly appreciate the time you have taken to review our work and provide detailed insights. Your comments help us a lot in refining our next version of the paper.
>
> We understand that you may have other commitments, but if you have any additional questions or require further clarification on any of our responses, we would be grateful for the opportunity to address them. **Your feedback is crucial to us, and we are eager to ensure that our revisions align with your expectations.**
>
> Please feel free to reach out at your convenience, as we would be more than happy to discuss any remaining concerns or provide further information.
>
> Thank you for your continued support.

---

### Author Rebuttal · Authors · 2024-08-07

Dear Reviewers,

We sincerely appreciate your time, consideration, and valuable comments, which have been instrumental in refining our work. If you have any further questions or concerns regarding our response or the current draft, please let us know. We are more than happy to discuss them in detail.

---

### Author Response · Authors · 2024-08-14
**Thank you for your review**

Dear reviewers,

We would like to express our sincere gratitude for your thoughtful and detailed reviews of my paper. Your feedback has been invaluable in helping us refine and strengthen our work. We truly appreciate the time and effort each of you has dedicated to providing constructive criticism and suggestions.

We will carefully apply revisions to the paper based on all the discussions and findings, ensuring that it meets the rigorous expectations of both the reviewers and the broader academic community.

We are also open to further discussions. Please feel free to let us know if you have any additional questions or concerns. We are happy to continue the dialogue with you.

Thank you once again for your review and support. We hope you have a great week.

---

### Decision · Program_Chairs · 2024-09-25

**Decision:**

Accept (poster)

**Comment:**

This paper carefully crafted a new method for explanation of predictions of graph regression neural networks. The method utilizes a novel objective based on information bottle neck theory and new mix-up framework; it is an add-on (post training) model agnostic method.

This paper is well written and easy to follow. Several reviewers praise the theoretically justified objective that was proposed. While experiments are a weak part of this paper, as mentioned by several reviewers,  synthetic datasets performance seems really convincing. The area of explainers for regression GNN is under-explored and will certainly benefit from this paper. Authors provide the code as an artifact, that will further increase the adoption of this work.
Furthermore, authors did an excellent job rebutting, resulting in several raised scores.

The paper should be strengthened by adding new experimental results on real world datasets.